# User Identity Linkage across Social Networks with the Enhancement of Knowledge Graph and Time Decay Function

**DOI:** 10.3390/e24111603

**Published:** 2022-11-04

**Authors:** Hao Gao, Yongqing Wang, Jiangli Shao, Huawei Shen, Xueqi Cheng

**Affiliations:** 1Data Intelligence System Research Center, Institute of Computing Technology, Chinese Academy of Sciences, Beijing 100190, China; 2University of Chinese Academy of Sciences, Beijing 100190, China; 3CAS Key Laboratory of Network Data Science and Technology, Institute of Computing Technology, Chinese Academy of Sciences, Beijing 100190, China

**Keywords:** user identity linkage, knowledge graph, named entity, time decay function, text matching

## Abstract

Users participate in multiple social networks for different services. User identity linkage aims to predict whether users across different social networks refer to the same person, and it has received significant attention for downstream tasks such as recommendation and user profiling. Recently, researchers proposed measuring the relevance of user-generated content to predict identity linkages of users. However, there are two challenging problems with existing content-based methods: first, barely considering the word similarities of texts is insufficient where the semantical correlations of named entities in the texts are ignored; second, most methods use time discretization technology, where the texts are divided into different time slices, resulting in failure of relevance modeling. To address these issues, we propose a user identity linkage model with the enhancement of a knowledge graph and continuous time decay functions that are designed for mitigating the influence of time discretization. Apart from modeling the correlations of the words, we extract the named entities in the texts and link them into the knowledge graph to capture the correlations of named entities. The semantics of texts are enhanced through the external knowledge of the named entities in the knowledge graph, and the similarity discrimination of the texts is also improved. Furthermore, we propose continuous time decay functions to capture the closeness of the posting time of texts instead of time discretization to avoid the matching error of texts. We conduct experiments on two real public datasets, and the experimental results show that the proposed method outperforms state-of-the-art methods.

## 1. Introduction

With the development of social networks, users usually own multiple social network accounts for different purposes. In particular, a report (https://backlinko.com/social-media-users) (accessed on 3 October 2022) has shown that the average number of social media accounts of one person in 2020 is 8.4. As illustrated in Figure 1, user identity linkage aims to predict whether the accounts across different social networks refer to the same person in reality, which supplements the data sources for downstream tasks by breaking the isolation of user information. For example, the sparse user information causes cold-start problems in recommendation systems, where users’ information is missing for recommendation. With the help of user identity linkage, the corresponding relationships of users across different social networks are constructed, and the information can be transferred from mature social networks to the target social networks for recommendations [1,2].

Exploring the correlations of content generated by users for predicting user identity linkage has become a hot topic in recent years. Based on the observations of user-generated content in different social networks, studies [3,4,5,6] claim that the behavior of users is consistent across social networks where they are prone to post **similar content** in **a close time period**. The key goal behind these methods is to measure the relevance of semantics of texts and the closeness of the posting time of users simultaneously.

Despite the success achieved by the above methods, there are two main problems in existing content-based methods: first, from the perspective of semantics of texts, existing methods only focus attention on modeling the similarities of words to predict the identity linkage of users, while they fail to explore the similarity of named entities hidden in the texts. Named entities are connected by relations in the knowledge graph, and the named entities introduce external knowledge as additional semantics of the texts. Therefore, the similarities of texts are enhanced through the knowledge graph, and the users with more semantically similar texts are more likely to be classified as the same user. Second, from the perspective of posting time of texts, existing methods discretize the timeline of the texts and model the similarity of texts in each time slice to capture the closeness of posting time. However, after discretization, the texts may fall into different time slices even if their posting time is close, thus the similarity of semantics cannot be measured, which misleads user identity linkage. We illustrate the problems in Figure 2.

This study addresses two research questions. The first is how to enhance the semantics of texts to predict user identity linkage. The texts posted by users contain many named entities, which are connected in the knowledge graph. The relationships of named entities supplement the correlations of users across different social networks, which helps to predict whether their identity is consistent. The second question is how to refrain from time discretization to capture the closeness of posting time. Time discretization causes missing matching in the texts, which leads to the failure of identity linkage prediction.

We propose a user identity linkage model with the **E**nhancement of **K**nowledge **G**raph (**EKG**), where the continuous time decay function is designed to capture the closeness of posting time. Apart from modeling the similarities of words in texts, we capture the similarities of named entities in the knowledge graph to enhance the semantical similarities of the texts. In addition, we introduce the continuous time decay functions to circumvent the failure of matching caused by time discretization. With the time decay function, the similarities of words or named entities are reweighted based on the posting time to capture the closeness of time.

In detail, we first extract the named entities in the texts and link them to the knowledge graph. To model the correlations of the semantics of texts, we construct the word and named entity similarity matrix and reweight the word and named entity similarity matrix based on the posting time through time decay functions. We then leverage the convolutional neural networks to learn similarity signals. Finally, an attention mechanism is applied to aggregate the word and named entity similarity signals to predict the user identity linkage. We evaluate the effectiveness of the proposed method on two real public datasets, namely, Foursquare–Twitter and Flickr–Yelp.

Our main contributions can be summarized as follows:We propose a novel user identity linkage method based on user-generated content, in which the semantics of texts posted by users are enhanced by introducing the additional correlations of named entities in a knowledge graph. To the best of our knowledge, this is the first time that named entities of a knowledge graph are considered in the user identity linkage task.We propose continuous time decay functions to capture closeness of posting time to circumvent the problem that similar content may fall into different time slices caused by time discretization.We conduct experiments on two real public datasets to demonstrate the effectiveness of our proposed method. Furthermore, we show the detailed ablation study to prove that named entity similarity modeling and the time decay functions significantly contribute to the results.

The rest of the paper is organized as follows: Section 2 introduces recent advances in user identity linkage methods; Section 3 presents the problem formulation for this task; Section 4 presents the proposed model, where similarities of words and named entities are captured to predict the identity of users; Section 5 describes the experimental data, evaluation methods, baseline models, and experimental results with analysis; Section 6 introduces the limitations of our study and future investigations; Finally, Section 7 summarizes this work.

## 2. Related Work

User identity linkage can be divided into three categories based on user information sources: profile-based methods [7,8,9,10], network-based methods [11,12,13,14,15], and behavior-based methods [16,17,18,19,20].

The basic assumption of the profile-based methods is that users may fill in similar registration information on different social networks. The main goal of these methods is to capture the similarities of the profiles of users. For example, Zafarani et al. [7] claimed that usernames are discriminative for predicting the identities of users, and they extract features about usernames to capture the similarity of users, including edit distance, uniqueness, typing patterns, etc. Mu et al. [21] proposed representing users in latent space via the profile features to predict the identities of users. Although these methods are effective and efficient, they ignore the fact that users may fill in the different information across social networks due to their writing habits, leading to the failure of identity linkage.

Another category of predicting user identity linkage is to leverage the network structure of users to capture the similarities of users [11,12,13,22]. The key idea is to measure the consistency of the local structure of the users to predict whether the user identity is the same. For example, Man et al. [12] proposed a two-stage model, which first learns the local network representations of the user, and then learns a mapping function to measure the distance between users across the network for prediction. Furthermore, Wang et al. [23] proposed learning the hash representation of users to speed up the matching process. Gao et al. [11] proposed using graph convolutional neural networks to model the cross-network local consistency structure of users to predict user identities. However, the friends of users across different social networks are not likely to be overlapping when the functionalities of social networks are different, resulting in a large gap in the network topology of users.

In recent years, the behavior-based methods have been widely studied [3,4,5,6,18,24], where the users’ check-ins or **generated contents** are modeled for predicting identity linkages of users. Feng et al. [18] proposed learning embeddings of the user’s check-in trajectory through Long Short Term Memory (LSTM), and employed an attention mechanism to weight the discriminative check-ins to capture the correlation of users. However, check-ins are sparse on non-geographic social networks, and the check-in-based methods may not work well. For example, only 8% of tweets have geolocation information on Twitter in our collected data. Liu et al. [25] divided the contents based on the time with different time slices and measure the similarities of users by the similarities of the topic distributions in the time slices. Srivastava et al. [4] explored text to extract the user’s writing habits, parts of speech, etc., to classify whether the user’s identity is consistent. However, users’ writing styles vary across social networks, and this method is sensitive to noise. Nie et al. [3] proposed dividing the posting time into several time slices and learning the dynamic topics of users from posted content to measure the similarities of users. Furthermore, Gao et al. [6] first discretized the texts into several slices based on posting time, and then measured the similarities of texts via the text matching process in each slice to capture the correlations of users. Despite the success of the above methods, they fail to capture deeper semantic interactions of the text. In addition, the discretization of texts may cause similar content to fall into different slices, leading to the failure of predicting identity linkages of users.

## 3. Problem Formulation

Without loss of generality, we assume that user identity linkages are predicted on two social networks, denoted as social network *A* and social network *B*, respectively. More platforms can be expanded through pairwise prediction of two social networks. For a user $u_{i}^{A}$ from social network *A* and a user $u_{j}^{B}$ from social network *B*, the content they post is denoted as $C_{i}^{A}$ and $C_{j}^{B}$, individually. The problem of user identity linkage is formulated as follows:(1)$$F\left(C_{i}^{A},C_{j}^{B}\right)=\left\{\begin{array}{ccc} 1 & \; & u_{i}^{A}\;and\;u_{j}^{B}\;are\;the\;same\;user.\\ 0 & \; & otherwise. \end{array}\right.$$
where *F* is the binary classification function to be learned to predict the identity linkages of users. The output of *F* is 1 if the users’ identities from different social networks are consistent, and 0 otherwise.

## 4. Model

Figure 3 shows the framework of our proposed model (denoted as **EKG**), which is composed of three modules: word and named entity representation, word matching and named entity matching, and prediction. In the representation module, we first extract the named entities through named entity recognition and link them into the knowledge graph to capture the semantics. To capture the interactions of words and named entities individually, we first initialize the representations of words and named entities and construct the matching matrices of words and named entities, where the similarities of semantics on the word and named entity level are obtained in the matrices. Users with the identity linkage are likely to post similar content in a close time period. Therefore, we introduce time decay functions to reweight the similarity matrices to capture the closeness of posting time. We apply the convolutional neural network to learn the similarity signals in the reweighted word and named entity matching matrices. Finally, the prediction module aggregates the learned word and entity similarity signals to predict whether the users from different social networks have the identity linkage. We will introduce the details of the modules in the following sections. Table 1 summarizes the symbols in the study.

### 4.1. Word and Named Entity Representation

To capture the semantic information hidden behind text words, we initialize word representations with unsupervised word representation learning algorithms. Word embedding learning algorithms, such as Word2Vec [26], GloVe [27], and BERT [28], are widely used in natural language processing tasks. Specifically, we initialize word vectors with GloVe, an algorithm that models the co-occurrence relationships between words appearing in the same window, and words with similar semantics have similar representations. For the texts $C_{i}^{A}$ posted by a user $u_{i}^{A}$, we first preprocess the texts and concatenate the words chronologically. We denote the concatenated words as $W^{i}=\left\{W_{1}^{i},W_{2}^{i},\ldots ,W_{M}^{i}\right\}$, where *M* is the total number of words posted by the user $u_{i}^{A}$, and the corresponding posting time is denoted as $T_{w}^{i}=\left\{T_{1}^{i},T_{2}^{i},\ldots ,T_{M}^{i}\right\}$. Similarly, for the user $u_{j}^{B}$, we denoted the concatenated words as $W^{j}=\left\{W_{1}^{j},W_{2}^{j},\ldots ,W_{N}^{j}\right\}$, and the corresponding posting time is $T_{w}^{j}=\left\{T_{1}^{j},T_{2}^{j},\ldots ,T_{N}^{j}\right\}$, where *N* is the total number of words posted by the user $u_{j}^{B}$. Furthermore, we initialize the word embeddings as $Z^{i}=\left\{z_{1}^{i},z_{2}^{i},\ldots ,z_{M}^{i}\right\}$ and $Z^{j}=\left\{z_{1}^{j},z_{2}^{j},\ldots ,z_{N}^{j}\right\}$, where $z_{m}^{i}$ and $z_{n}^{j}$ is the vector representations of the word $W_{m}^{i}$ and $W_{n}^{j}$, respectively.

Apart from the words in texts, capturing the correlations of named entities hidden in the texts enhances the semantics of user-generated content, which provides discriminative evidence for predicting the identity linkage of users. We extract named entities in the content and map them to the corresponding entities in the knowledge graph for introducing external knowledge. We denote the mapped entities in the knowledge graph of user $u_{i}^{A}$ and $u_{j}^{B}$ as $E^{i}=\left\{E_{1}^{i},E_{2}^{i},\ldots ,E_{P}^{i}\right\}$ and $E^{j}=\left\{E_{1}^{j},E_{2}^{j},\ldots ,E_{Q}^{j}\right\}$, and the corresponding posting times are denoted as $T_{e}^{i}=\left\{T_{1}^{i},T_{2}^{i},\ldots ,T_{P}^{i}\right\}$ and $T_{e}^{j}=\left\{T_{1}^{j},T_{2}^{j},\ldots ,T_{Q}^{j}\right\}$, where *P* and *Q* are the numbers of the entities posted by user $u_{i}^{A}$ and $u_{j}^{B}$. After obtaining the entities, we initialize the embeddings of the entities. Different from the sequential structure of words in texts, the entities are connected with relationships. We leverage TransE [29] to initialize the entity representations. TransE is a deep learning method that learns the embeddings of named entities by capturing the relationships between the entities, of which the assumption is that the tail entity can be represented by the head entity and their relationships. We denote the entity representations of named entities $E^{i}$ and $E^{j}$ as $H^{i}=\left\{h_{1}^{i},h_{2}^{i},\ldots ,h_{P}^{i}\right\}$ and $H^{j}=\left\{h_{1}^{j},h_{2}^{j},\ldots ,h_{Q}^{j}\right\}$, where $h_{p}^{i}$ and $h_{q}^{j}$ are the vector representations of named entities $E_{p}^{i}$ and $E_{q}^{j}$, respectively.

### 4.2. Word and Named Entity Matching

The basic assumption is that users with the same identities from two social networks behave similarly. In other words, users are prone to posting similar contents in close time periods if their identities are the same. Thus, the key goal is to capture the correlations of semantics hidden in the texts and the closeness of posting time. We construct the word and named entity similarity matrices based on the interactions of words and named entities, where the relevance of the semantics is captured in the matrices. To measure the closeness of posting time, we introduce time decay functions to reweight the matrices based on the posting time. When semantics are relevant and the posting time is close, a higher value is obtained, which indicates the similar behaviors of users, and users are more likely to have identity linkage. Finally, we learn the matching signals of the word and named entity similarity matrices through convolutional neural networks.

#### 4.2.1. Word Matching

We first construct the word similarity matrix $M^{w}$, in which each element stands for the cosine similarities of words in the text posted by users $u_{i}^{A}$ and $u_{j}^{B}$, i.e.,
(2)$$M_{m,n}^{w}=\frac{z_{m}^{i}\cdot z_{n}^{j}}{\left|\right|z_{m}^{i}\left||*|\right|z_{n}^{j}\left|\right|}$$
where $M_{m,n}^{w}$ is the score of the *m*th row and *n*th column in word similarity matrix $M^{w}$, $z_{m}^{i}$ is the embeddings of the *m*th word in the content posted by the user $u_{i}^{A}$, and $z_{n}^{j}$ is the embeddings of the *n*th word in the content posted by user $u_{j}^{B}$. To mitigate the problem that the scores of dissimilar words are accumulated as higher values that misleads the matching, we filter the word similarity matrix, where scores less than the threshold *r* are discarded, i.e., set as 0 in our scenario.

To capture the closeness of posting time, we propose to attenuate the word similarity matrix by the continuous time decay functions instead of time discretization, where the similarities of words with a larger time gap are penalized. Specifically, for the word sets $W^{i}$ and $W^{j}$ with the corresponding posting time $T_{w}^{i}=\left\{T_{1}^{i},T_{2}^{i},\ldots ,T_{M}^{i}\right\}$ and $T_{w}^{j}=\left\{T_{1}^{w},T_{2}^{w},\ldots ,T_{N}^{w}\right\}$, we first introduce the time difference matrix $\Delta T^{w}$, of which each element is defined as
(3)$$\Delta T_{m,n}^{w}=d(|T_{m}^{i}-T_{n}^{j}\left|\right)$$
where $\Delta T_{m,n}^{w}$ is the value of *m*th row and *n*th column in time difference matrix $\Delta T^{w}$, $T_{m}^{i}$ and $T_{n}^{j}$ are the corresponding posting time of word $W_{m}^{i}$ and $W_{n}^{j}$, and $d\left(\cdot \right)$ is the time decay function. As the time gap increases, the corresponding value in $\Delta T^{w}$ will decrease. We define three time decay functions, namely step decay function, linear decay function, and exponential decay function, which are formulated as follows:(4)$$d_{1}\left(x\right)=sign\left(x-t_{0}\right),t_{0}>0$$
(5)$$d_{2}\left(x\right)=\max\left(wx+b,0\right),w<0,b=1$$
(6)$$d_{3}\left(x\right)=e^{\left(-\alpha x\right)},\alpha >0$$
where $sign\left(\cdot \right)$ is an indicator function in Equation (Equation 4), and we define that the output is 1 if the *x* is less than $t_{0}$, and 0 otherwise. The time decay functions are monotonically decreasing, and the max value is reached as 1 when the time gap (i.e., the input *x*) is 0. The step decay function assumes that the word similarities remain consistent until reaching the threshold $t_{0}$. Otherwise, the word similarities are discarded. Compared to the step decay function, the value decreases linearly as the time gap grows in the linear decay function. Furthermore, in the exponential decay function, the value decreases rapidly at first, and decreases slowly as the time gap grows. Figure 4 illustrates three time decay functions.

The key idea behind the time decay function is to reweight the word similarity matrix based on the difference of the posting time of the words. The larger time gap indicates less importance of the word similarities for predicting the identity linkage of users. We penalize the word similarity matrix $M^{w}$ by the time difference matrix $\Delta T^{w}$ as follows:(7)$${\hat{M}}^{w}=M^{w}⊙\Delta T^{w}$$
where ⊙ is the Hadamard product of the matrix, i.e., the corresponding elements of the matrices are multiplied. After penalization, the similarity scores of words decrease when their posting time is far from one another. As a result, if and only if the words are similar and the posting time is close, the scores in ${\hat{M}}^{w}$ are high, which indicates that the identities of users may be the same.

#### 4.2.2. Named Entity Matching

Similarly, we now measure the similarities of named entities in users’ posts. We first introduce the named entity similarity matrix. For the representations of named entities $H^{i}=\left\{h_{1}^{i},h_{2}^{i},\ldots ,h_{P}^{i}\right\}$ and $H^{j}=\left\{h_{1}^{j},h_{2}^{j},\ldots ,h_{Q}^{j}\right\}$ of users $u_{i}^{A}$ and $u_{j}^{B}$, the named entity similarity matrix is denoted $M^{e}$, where each element is defined as follows:(8)$$M_{p,q}^{e}=\frac{h_{p}^{i}\cdot h_{q}^{j}}{\left|\right|h_{p}^{i}\left||*|\right|h_{q}^{j}\left|\right|}$$
where $M_{p,q}^{e}$ is score of the *p*th row and *q*th column in named entity similarity matrix $M^{e}$, $h_{p}^{i}$ is the embeddings of the *p*th named entity of user $u_{i}^{A}$, and $h_{q}^{j}$ is the embeddings of *q*th named entity of user $u_{j}^{B}$. Similarly, we filter the named entity similarity matrix by threshold *r*. We reweight the named entity similarity matrix $M^{e}$ by the posting time of named entities. The time difference matrix of named entity $\Delta T^{e}$ is defined as follows:(9)$$\Delta T_{p,q}^{e}=d(|T_{p}^{i}-T_{q}^{j}\left|\right)$$
where $d\left(\cdot \right)$ is the time decay function as shown in Equations (Equation 4)–(Equation 6); $T_{p}^{i}$ and $T_{q}^{j}$ are the corresponding posting time of named entities $E_{p}^{i}$ and $E_{q}^{j}$. We penalize the named entity similarity matrix by time difference matrix as follows:(10)$${\hat{M}}^{e}=M^{e}⊙\Delta T^{e}$$
The score in ${\hat{M}}^{e}$ is higher when the semantics of named entities are relevant and the corresponding posting time is close. The reweighted named entity similarity matrix enhances the relations of contents semantics of users across social networks, providing evidence for user identity linkage.

#### 4.2.3. Matching Signal Extraction

Inspired by the field of computer vision where the convolutional neural network (CNN) extracts pixels, edges, and patterns features of images, we apply CNN on reweighted word and named entity similarity matrix ${\hat{M}}^{w}$ and ${\hat{M}}^{e}$ to capture higher-level matching patterns analogously. In detail, we use two two-layer convolutional networks to learn the matching signals of word and named entity, respectively. The convolutional and pooling operators are defined as follows:(11)$$\begin{array}{cc} \; & x_{\left(k\right)}=\sigma \left(\sum_{i}w_{\left(k\right)}^{i}🟉x_{\left(k-1\right)}+b_{\left(k-1\right)}\right)\\ \; & x_{\left(k\right)}=MaxPooling\left(x_{\left(k\right)}\right) \end{array}$$
where 🟉 is the convolution operator, and MaxPooling outputs the max value of the feature map, $x_{\left(k\right)}$ is the feature map after *k*-layer convolution networks, $w_{\left(k\right)}^{i}$ and $b_{\left(k-1\right)}$ are the weights of convolution kernels and the bias, and σ is activation function, where ReLU [30] is adopted to avoid the problem of gradient disappearance in our scenario. We initialize the input of the convolutional neural networks (i.e., $x_{0}$) as the reweighted word similarity matrix ${\hat{M}}^{w}$ and reweighted named entity similarity matrix ${\hat{M}}^{e}$, respectively. Moreover, we feed the final feature map through a fully connected layer for global similarity signal capturing and dimensionality reduction. Finally, the word and named entity matching signals are learned, denoted as $x_{w}$ and $x_{e}$, respectively.

### 4.3. Prediction Module

To predict user identity linkage, we aggregate word and named entity matching signals. It is straightforward to concatenate or add the word and entity matching signals $x_{w}$ and $x_{e}$, and then classify them through a fully connected layer. However, both the concatenation and adding assume that the word and named entity contribute the same to our task, while their importance is different. Thus, we employ an attention mechanism to balance the word and named entity matching signals, allowing our model to focus more on the parts that are discriminative to user identity linkage task. In detail, for the word matching signal $x_{w}$ and named entity matching signal $x_{e}$, the attention mechanism is formalized as follows:(12)$$u_{i}=\tanh\left(W_{a}x_{i}+b_{a}\right)$$
(13)$$\alpha_{i}=\frac{\exp\left(u_{i}^{T}u_{w}\right)}{\sum_{k}\exp\left(u_{k}^{T}u_{w}\right)}$$
(14)$${\hat{x}}_{i}=\alpha_{i}u_{i}$$
where $x_{i}$ is the representations of matching signals $x_{w}$ or $x_{e}$, $W_{a}$ and $b_{a}$ are the parameters to be learned, ${\hat{x}}_{i}$ is the weighted similarity representations, and $u_{w}$ is a learned parameter that weights the importance of the similarity representations. We denote the weighted matching signals as ${\hat{x}}_{w}$ and ${\hat{x}}_{e}$. To predict whether user identities are consistent, we concatenate the ${\hat{x}}_{w}$ and ${\hat{x}}_{e}$, and feed them into a fully connected layer, i.e.,
(15)$$o=f\left(W_{d}\left[{\hat{x}}_{w};{\hat{x}}_{e}\right]+b_{d}\right)$$
where $W_{d}$ and $b_{d}$ are the weights and bias of the fully connected network, *f* is the sigmoid function, which normalizes the predicted results, and *o* is the final output of the model. We employ cross entropy loss to train the model, i.e.,
(16)$$L=-\frac{1}{N}\sum_{i=1}^{N}y_{i}*log\left(o_{i}\right)+\left(1-y_{i}\right)*log\left(1-o_{i}\right)$$
where *N* is the number of the samples, and $y_{i}$ and $o_{i}$ are the ground-truth and the prediction result of *i*th sample, respectively.

### 4.4. Overall Procedures

To elaborate on the overall procedures of our proposed model, we describe the detailed calculation of the sample in Figure 3 and encapsulate the algorithm in Algorithm 1. As shown in Figure 3, users from two social networks post texts, in which the words and the named entities are included after preprocessing (the details of preprocessing are described in Section 5.1). We first initialize the representations of words and named entities. Then we calculate the word and named entity similarity matrices $M^{w}$ and $M^{e}$ based on Equations (Equation 2) and (Equation 8), respectively. In addition, we calculate the word and named entity time difference matrices $\Delta T^{w}$ and $\Delta T^{e}$ based on Equations (Equation 3) and (Equation 9), where the time decay functions are applied for capturing the closeness of the posting time. Then the similarity matrices and time difference matrices of word and named entity are combined through element-wise production to simultaneously obtain the correlations of semantics of texts and posting time, which is described in Equations (Equation 7) and (Equation 10). To learn the matching signals, we apply convolution neural networks to extract features, as shown in Equation (Equation 11). Finally, we leverage an attention mechanism to aggregate the word and named entity matching signals to enhance the semantics correlations of texts. To train the model, we adopt cross entropy loss and backward propagation to optimize the parameters.
**Algorithm 1:** Overall procedures of proposed model (**EKG**)
**Input**: Content CiA posted by user uiA; Content CjB posted by user ujB; Filtering threshold *r*;
**Output**: The prediction result *o*;**1**Preprocess the content CiA and CjB;// See Section 5.1; **2**Initialize the representations of word and named entity;// Calculate the word and named entity similarity matrices Mw and Me**3**Mm,nw←zmi·znj||zmi||∗||znj||;**4**Mp,qe←hpi·hqj||hpi||∗||hqj||**5**Filtering Mw and Me where the scores are less than threshold *r*;// Calculate the word and named entity time differences matrices ΔTw and ΔTe**6**ΔTm,nw←d(|Tmi−Tnj|);**7**ΔTp,qe←d(|Tpi−Tqj|); // Reweight word and named entity similarity matrices based on posting time;**8**M^w=Mw⊙ΔTw;**9**M^e=Me⊙ΔTe;**10**Extract word and named entity matching signals xw and xe based on Equation (Equation 11);**11**Aggregate the word and named entity matching signals through an attention mechanism based on Equations (Equation 12)–(Equation 14);**12**
Predict the probability *o* that the user identities are consistent based on Equation (Equation 15);**13****return** *o*

## 5. Experiment

In this section, we describe our experiment settings and results. We released our code publicly https://github.com/goleey/EKG (accessed on 3 October 2022).

### 5.1. Dataset Collection and Preprocessing

**Dataset:** To demonstrate the effectiveness of our proposed model, we conduct experiments on two real public datasets, namely, Foursquare–Twitter [9,31] and Flickr–Yelp [32]. The statistics of the datasets are shown in Table 2. For privacy concerns, all the IDs of the users are mapped to the anonymized IDs in the collected datasets.

**Foursquare–Twitter:** Foursquare is a location-based social network where users post their comments on the place that they visit. Twitter is a microblogging social network where users communicate with each other. As described in the literature [31], they first crawled the users and their posts by using the APIs of Foursquare and Twitter. Next, the ground-truth is obtained through the hyperlinks that users released on their homepages publicly on Foursquare. The dataset consists of 5392 Foursquare users and 5223 Twitter users with their posting content and the corresponding check-ins. Furthermore, the number of users with the same identities is 3388.**Flickr–Yelp:** Flickr is a photo-sharing platform where users share photos with descriptions. Yelp is a location-based social network where users comment on check-in places. As the literature [32] states, they first collected the users by exploiting the “Friend Finder” mechanism present on many social networking sites. After that, they used an existing list of e-mails (The list is collected from an earlier study by colleagues analyzing email spam. The local IRB approved collection and usage). to check if the e-mails correspond to the crawled results to obtain the ground-truth. Finally, they crawled the posts of the users via APIs of the social networks. We keep the users if the contents are posted within 3 months on two social networks. The dataset consists of 735 users on both Flickr and Yelp. In addition, the number of users with the same identities is 735 as well.

**Text Preprocessing:** The content posted by users on social networks are noisy due to colloquialization, symbols, emojis, etc. We first preprocess the texts by removing the above symbols and stop words via the rules. In detail, the symbols such as comma, period, ellipsis are removed, and stop words are removed by regex matching by the stop words list, which are collected by the NLTK (https://github.com/nltk/nltk).(accessed on 3 October 2022) In addition, the words have different morphologies, and the words with different forms may refer to the same semantics. To avoid the inconsistent morphology problem, we normalize the words by lemmatization. The lemmatization is implemented based on WrodNet (https://wordnet.princeton.edu/) (accessed on 3 October 2022) which is a database that contains the relationships of words. After lemmatization, the words with different forms are transferred to the same original forms. For example, the word “ate” is transferred to the word “eat”.

**Named Entity Recognition and Linking:** Apart from capturing the word relevance in texts, the correlations of named entities hidden in the texts enhance the semantics of texts posted by users across different social networks, which provides discriminative evidence for predicting the identity linkages of users. We start with extracting entities from the content of users, which is formulated as a named entity recognition task (NER). NER aims to seek the entities existing in texts and classify them into categories, including organization, location, quantity, etc. We adopt the transition-based chunking model proposed by Lample [33], which uses a stack structure to incrementally construct the segment as input, and then apply the stacked LSTM [34] to label the structure. Specifically, not all types of entities are valid in our task, and the named entities related to users’ behaviors are preserved, including organization, person, and location. After extracting entity mentions from the texts, we now map them to the corresponding entities in the knowledge graph for further introducing the external knowledge. Mapping the entity mentions to the entity in the knowledge graph is a fundamental task, which is widely studied. To map the entity mentions, we employ search engine-based entity linking methods [35,36,37,38], which leverage the whole web information for entity retrieval via search engines. Inspired by Han et al. [35], we submit the entity mentions together with their contexts into Wikipedia (https://www.wikipedia.org/) (accessed on 3 October 2022) to retrieve the corresponding entities for disambiguation.

### 5.2. Baselines

To compare the performance of the proposed method, we implement the following baselines that capture user behaviors to predict the user identity linkage, where content or check-in of the users across social networks are modeled. Four categories of baseline methods are used, including feature-based methods, topic-based methods, semantic-based methods, and check-in-based methods. In addition, we ensure that the text preprocessing in baselines is the same as our proposed methods for a fair comparison.

Feature-based methods:**WAI** [4] (Short for the title “**W**ords **A**re **I**mportant: A Textual Content Based Identity Resolution Scheme Across Multiple Online Social Networks”) extracts the part-of-speech, symbols, expressions, and high-frequency words of users’ content and then trains classifiers to predict user identities. Furthermore, WAI analyzes the importance of different features.**LSIF** [5] (Short for the title “User Identity Linkage in Social Media Using **L**inguistic and **S**ocial **I**nteraction **F**eatures”) extracts not only text features of users, but also network features. The text features include the number of texts posted by users, topics, writing styles, etc. For a fair comparison, we only adopt the textual features of this work.Topic-based methods:**UBL** [39] (Short for the title “Matching **U**ser Accounts Across Social Networks **B**ased on **L**DA Model”) proposes to learn the topics of contents through author topic model and dynamic topic model. The assumption is that user’s topics are similar if their identities are the same. Finally, Hellinger distance and residual sum of squares are applied to measure the similarities of the topics to predict user identity linkages.**DCIM** [3] (Short for the title “Identifying Users Across Social Networks Based on **D**ynamic **C**ore **I**nterests”) predicts user identities by modeling their temporal topic distribution of the contents. DCIM first divides the text into different slices in chronological order and then uses LDA to learn the core topic distribution of the text within the slices. Finally, user pairs with similar topics are classified as the same identities.Semantic-based method:**UGCLink** [6] (Short for the title “**UGCLink**: User Identity Linkage by Modeling User Generated Contents with Knowledge Distillation”) is the closest work to our proposed model. UGCLink proposes to discretize text by time and use convolutional neural networks to model word, phrase, and sentence-level similarity signals within each time slice. In addition, the model uses knowledge distillation to make the semantic representations of the learned words have geographic location information. Finally, a binary classification model is used to predict user identity.Check-in-based method:**Dplink** [18] (Short for the title “**DPLink**: User Identity Linkage via Deep Neural Network From Heterogeneous Mobility Data”) learns the trajectory representations of users, and users with similar representations might have the same identity. Specifically, Dplink designs an encoding model where spatio-temporal information of users is mapped into the latent space.

### 5.3. Evaluation Metrics

To compare the effectiveness of baselines and our proposed method, we apply various evaluation metrics, including precision, recall, F1 score, and AUC score. Precision, recall, and F1 score are defined as
(17)$$Precision=\frac{TP}{TP+FP}$$
(18)$$Recall=\frac{TP}{TP+FN}$$
(19)$$F1=\frac{2*Precision*Recall}{Precision+Recall}$$
where *TP*, *FP*, *FN*, and *TN* are the true positive, false positive, false negative, and true negative in the confusion matrix. Furthermore, we adopt the AUC score to evaluate the effectiveness, which is the area under the receiver operating characteristics (ROC) curve. A higher AUC score indicates that the probability of ranking a randomly chosen positive sample is higher than a random negative one.

### 5.4. Environment and Hyperparameter Settings

All the experiments are performed on a Linux machine with 2.7 GHz Intel cores and 128G RAM. We implemented the model by PyTorch 1.1. The model is optimized with the Adam optimizer. In addition, the batch size is 32, and the embedding dimensions of words and entities are both set to 50. We set the layers of convolution networks as 2, where the sizes of kernels are 3 × 3 × 5 and 5 × 5 × 5, respectively. All the parameters are initialized by the Xavier Initialization [40] method. We conduct the experiments with three proposed time decay functions, and the exponential decay function outperforms the others. We report the results under the exponential decay function. The time decay factor α is experimentally set to 0.0006, and the basic unit of input is the hour. We randomly sample negative samples and the ratio of negative examples to positive examples is 1:1. After that, the dataset is randomly divided into the training set, validation set, and test set with a ratio of 7:1:2 for analysis. We ensure that the ratio of positive and negative samples is 1:1 in each set.

### 5.5. Experiment Results

Table 3 presents the comparison of experimental results of the baseline models and our proposed model (The result of Dplink on Flickr–Yelp is not reported due to the check-ins not being collected in this dataset.) We can draw the following conclusions from the table.

Our proposed method EKG outperforms all the baseline models. In addition, the methods of modeling semantic information (UGCLink, our proposed EKG model) are more effective than the methods of trivial feature extraction (WAI, LSIF). The feature extraction methods are trained by extracting features such as expressions and parts of speech from the texts. However, these features are often noisy due to users’ posting habits, leading to the misclassification of user identity linkage.

Compared with topic-based methods (UBL, DCIM), which learn the topics of the user’s text to capture the user’s similarity, our proposed method models the interactions of the representations of words and named entities individually. The topic-based methods fail to explore the correlations of the semantics of texts. The users are predicted with the identity linkage only if the topic distributions are similar. However, users on different social networks may not post words that are exactly the same.

The recall of DCIM is higher than our proposed model in Foursquare–Twitter dataset, whereas the precision of DCIM is much worse. We further analyze the core topics extracted by DCIM in Twitter, and we found that the topics mostly include the oral words that are often used by most users. Thus the recall score is high; however, the precision score is low.

We notice that the precision and recall score differ in the two datasets for all methods. The precision score is relatively higher than recall in Foursquare–Twitter, whereas the opposite results are reported in Flickr–Yelp. The reason is that the distributions are different in these two datasets, and the styles of content on Twitter are different from that in Flickr.

UGCLink is a semantic-based model which discretizes the content of users by time and fails to explore the entity similarity existing in the text. Compared to UGCLink, our proposed method explicitly introduces the time decay function and entity correlation modeling, which is proven to be effective via the experiment results. Specifically, our proposed method additionally constructs the named entity similarity matrix to enhance the semantics relevance of texts, which is beneficial for exploring similar behaviors of users across different social networks.

Dplink is a check-in-based model that predicts user identity based solely on the user’s check-in information. However, the check-in data on Twitter are sparse, and most tweets are not associated with locations. Thus, it is hard to apply Dplink when social networks are not location-based. Our proposed method discards the check-in information. Instead, we propose to learn the correlations of words and named entities in texts to explore the behaviors of users to predict their identity linkages.

Our proposed method produces superior results compared to baselines, due to achieving two key goals in content-based user identity linkage, i.e., external knowledge introduced from a knowledge graph and the continuous time decay functions. We will discuss the effectiveness of these two improvements in what follows.

### 5.6. Ablation Study

To demonstrate the effectiveness of the time decay function, word matching module, and named entity matching module, we conduct an ablation study experiment by removing or replacing each part of our model on two datasets, as shown in Table 4 and Table 5. We design three variants of EKG, denoted as EKG-W, EKG-D, and EKG-E. EKG-W only retains the similarity modeling of words and removes the similarity modeling of named entities. In contrast, EKG-E only retains the similarity modeling of entities while removing the word similarity modeling. Furthermore, to evaluate the effectiveness of the time decay function, we replace the time function with the time discretization on the basis of EKG-W, where the texts are divided into several time slices. We denote it as EKG-D.

It can be observed that either removing or replacing the modules degrades the performance compared with our proposed model. Compared to EKG, we remove the entity similarity modeling in EKG-W, which degrades AUC by 5% on the Foursquare–Twitter dataset. The reason is that extracted named entities (e.g., location, person) enhance the semantics of texts, where the similarities of texts are more discriminative when predicting user identity linkages. Capturing the similarity of named entities promotes the user identity linkage task.

On the basis of EKG-W, we design EKG-D, where the time decay function is replaced with the discretization of texts by posting time. We can observe that the performance drops. When dividing the texts into slices by time, the texts with similar semantics may fall into different slices. Thus the similarities of these texts are not modeled, leading to the failure of user identity linkage.

Moreover, to demonstrate the effectiveness of word similarity modeling, we remove the word similarity modeling module, while only keeping the named entity similarity modeling. It is observed that the performance of EKG-E drops the most. The reason is that not all the texts in social networks consist of named entities. It is not sufficient to predict user identity linkage only based on the named entities due to the sparseness.

### 5.7. Hyperparameter Analysis

We analyze three main hyperparameters in our model on two datasets, e.g., different time decay functions, the threshold of filtering the word and named entity similarity matrix, and the decay factor in an exponential time decay function.

#### 5.7.1. Time Decay Functions

We propose three time decay functions to penalize the word and named entity similarity matrix. Figure 5 and Figure 6 show the performance of three decay functions on two datasets.

It can be seen that the exponential decay function achieves the best AUC score, followed by step decay, and finally linear decay. Compared with the other two decay functions, the gradient of the exponential decay function is large at first and tends to decrease as the time gap becomes larger. The exponential decay function smoothly captures the similarity of users’ interests over a long period of time, which is suitable for describing the dynamics of users’ posting content. The step function assumes that the similarity of texts exceeding a certain time threshold is 0 to focus the similarity matrix on the scores that are in a close time period. In addition, the linear function assumes that the similarity scores decrease in a linear way. Compared to the exponential decay function, the gradient is invariable as the time gap grows. However, as the time gap becomes larger, the score of the linear function is still higher than that of the exponential function, which may introduce noise to the similarity matrix. In general, we finally adopt the exponential decay function as the weighting function for the word and named entity similarity matrix.

#### 5.7.2. Threshold *r*

After obtaining the similarity matrices of the word and named entity, we apply a threshold *r* to discard the small scores to avoid the cumulative effect of small scores. For simplicity, the filter threshold is set to be the same value for the word and named entity similarity matrix. Figure 7 and Figure 8 show the results of different thresholds on two datasets.

When the threshold is low, most dissimilar words or named entities are kept in the similarity matrices, and the matching signals are accumulated as a high score, which misleads user identity linkage. As the threshold increases, the performance of classification improves, which indicates the effectiveness of our proposed filtering method. However, when the threshold reaches 1.0, that is, only the same words or named entities are retained, the performance drops. The reason is that it ignores the semantic similarities of words or named entities and classifies user identities strictly according to whether the words or entities are consistent. However, users seldom publish the exact same words on different social networks.

#### 5.7.3. Decay Factor α

To explore the effectiveness of different α in an exponential time decay function (see Equation (Equation 6)), we further explore the performance of our proposed model while α grows. Figure 9 and Figure 10 show the results of different α on two datasets.

As α increases, the performance first increases and then drops. When the time gap is the same, larger α indicates that the exponential time decay function outputs smaller values, which degrades the word and named entity similarity matrix more. Thus, the similarities of words or named entities with close posting time will decrease, and the texts with similar semantics and close posting time become irrelevant, leading to the failure of predicting user identity linkage. In contrast, similar texts with longer posting time gaps still output larger matching signals while α is small, where dissimilar texts with longer time periods may be considered to capture semantic relevance, introducing noises to predict the linkage of user identity.

## 6. Limitations and Future Investigations

The method in this study proposes to enhance the semantical similarities of texts with the knowledge graph, and continuous time decay functions are designed to avoid the failure of matching introduced by time discretization. We observe that the method achieves better performance compared to baselines. However, there are still some limitations of the study.

First, some users may not post similar texts across different social networks. The existing literature, including ours, which captures the similarity of texts posted by users, may fail in this scenario. To mitigate the above problem, future study should be undertaken to explore the additional information of users, such as attributes or check-ins, to model the correlations of users across different social networks.

Second, the ratio of the sampling negative users is set as 1:1 in our experiment. However, the number of negative users is more than 1 in practice. As a result, introducing more negative users will lead to imbalance classes, which makes it hard to train the model. Thus, a future direction can be learning the classifier under the imbalance settings of samples to improve the effectiveness of the user identity linkage task.

Third, the parameters of time decay functions are set to the same value for all users. However, different users may have different habits when they post the texts. For example, some users post texts after a long time gap across different social networks, while some users post texts on one social network once they post them on the other social network. An improvement could be learning the different parameters of time decay functions for different users based on their habits.

It is anticipated that introducing the additional information, sampling more negative users, or learning different parameters of time decay functions for different users will promote the results and achieve more accurate prediction results. We will further investigate these problems in future studies.

## 7. Conclusions

This study proposes a user identity linkage model with the enhancement of the knowledge graph, where a continuous time decay function is designed for capturing the closeness of posting time. Two main problems are proposed in this study. The first problem is that modeling the semantics of words is not sufficient for predicting user identity linkage. Moreover, dividing the texts into different time slices will lead to the failure of relevance modeling. To solve the above problems, apart from word similarity modeling, the proposed model extracts the named entities in the content and links them into a knowledge graph for enhancing the semantic relevance of texts. Furthermore, to solve the problem that similar words or entities may fall into different time slices after time discretization, continuous time decay functions are designed to reweight the word and named entity similarity matrix. Finally, the proposed model achieves more accurate results on the real public dataset compared to state-of-the-art baselines.

We limit our study to the users who post similar texts with close posting time. However, some users may not post similar texts across different social networks. Consequently, future work is directed to exploring the texts together with the check-in data to jointly capture the similarity of user behaviors to further improve the accuracy of user identity linkage.

## Figures and Tables

**Figure 1 entropy-24-01603-f001:**
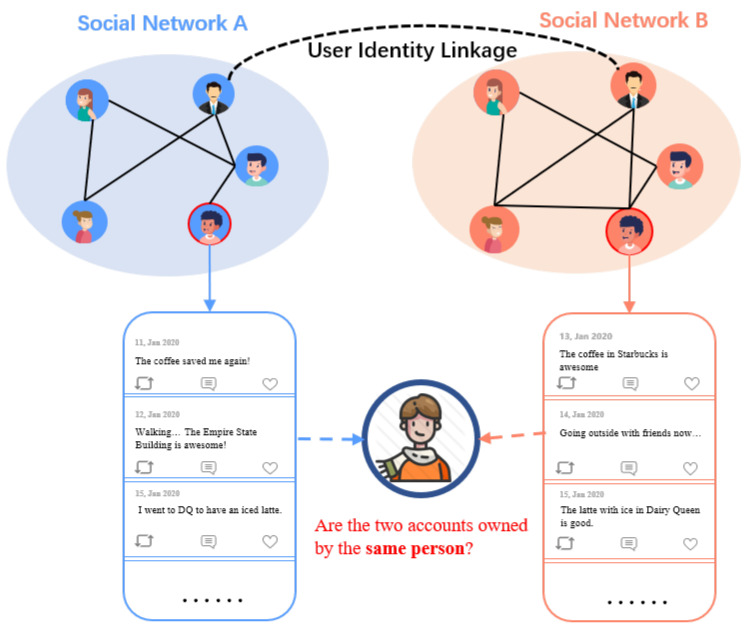
The illustration of user identity linkage task. The black dotted line (on the top) is the observed identity linkage between users, which indicates that the owner of two accounts is the same person. User identity linkage tasks aim to predict whether the two accounts from different social networks are owned by the same person.

**Figure 2 entropy-24-01603-f002:**
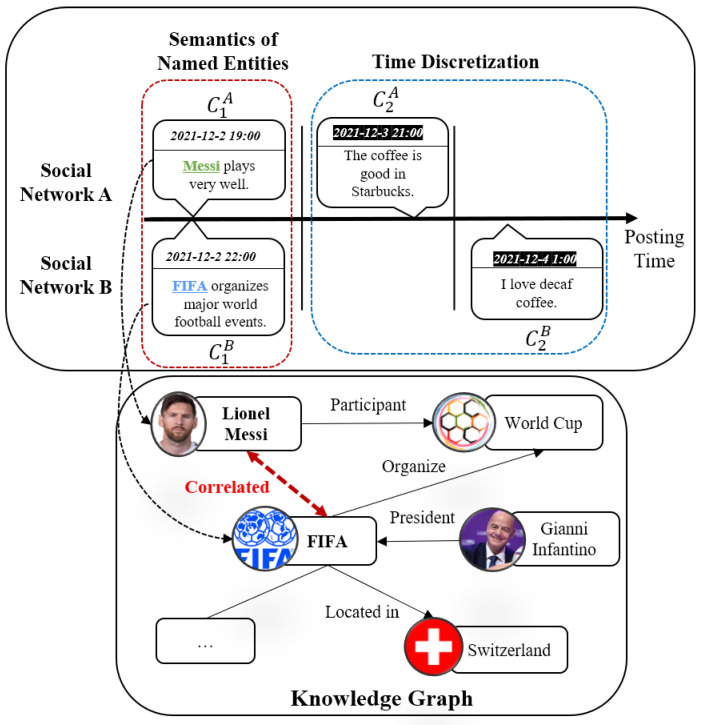
The example of problems in user identity linkage. The upper subfigure is the user-generated content across social networks A and B, where the vertical bar divides the content based on the posting time. The lower subfigure is the knowledge graph. The content of $C_{1}^{A}$ and $C_{1}^{B}$ contain the named entities “Messi” and “FIFA”, individually (in the red box). These two entities are closely related in the knowledge graph, and capturing the relations of entities enhances the similarities of semantics in content $C_{1}^{A}$ and $C_{1}^{B}$, which indicates the users are likely to be the same person. Furthermore, the content of $C_{2}^{A}$ and $C_{2}^{B}$ are similar and their posting time is close (in the blue box).

**Figure 3 entropy-24-01603-f003:**
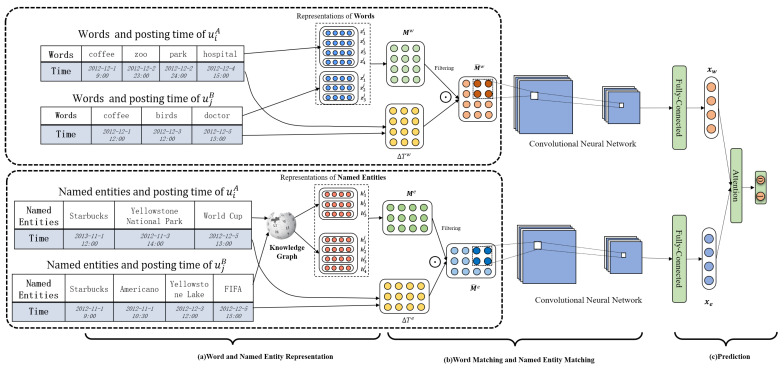
The framework of our proposed method: $u_{i}^{A}$ and $u_{j}^{B}$ are two users from different social networks, and their words and named entities with the posting time are shown in the table on the left. (**a**) For the words (on the top), the representations of words are initialized, and the similarity matrix based on the pairwise similarities of the words posted by user $u_{i}^{A}$ and $u_{j}^{B}$ are constructed. Next, the time difference matrix based on the posting time is calculated and used for reweighting the word similarity matrix by element-wise production. For the named entity (on the bottom), the named entity is first mapped to the knowledge graph for disambiguation and the representations are initialized for capturing the correlations in knowledge graph. Similarly, named entity matching matrix and time difference matrix are constructed, and the matching matrix is reweighted by the time difference matrix. (**b**) The convolution neural networks are applied for extracting the word and named entity matching signals. (**c**) The word and named entity matching signals are aggregated through an attention mechanism for final prediction.

**Figure 4 entropy-24-01603-f004:**
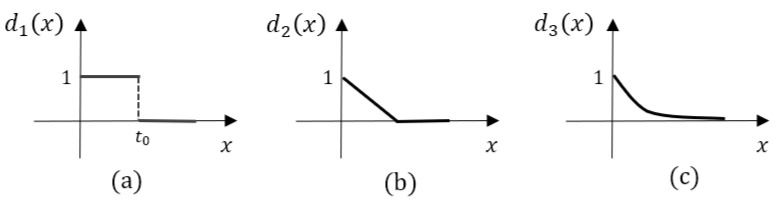
Three time decay functions. The horizontal coordinate is the time gap, and the vertical coordinate is the decay value: (**a**) step decay function; (**b**) linear decay function; (**c**) exponential decay function.

**Figure 5 entropy-24-01603-f005:**
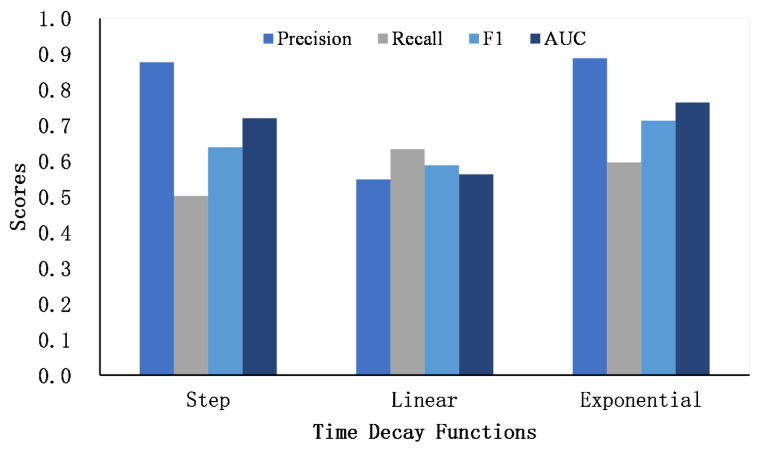
The comparison of time decay function on Foursquare–Twitter.

**Figure 6 entropy-24-01603-f006:**
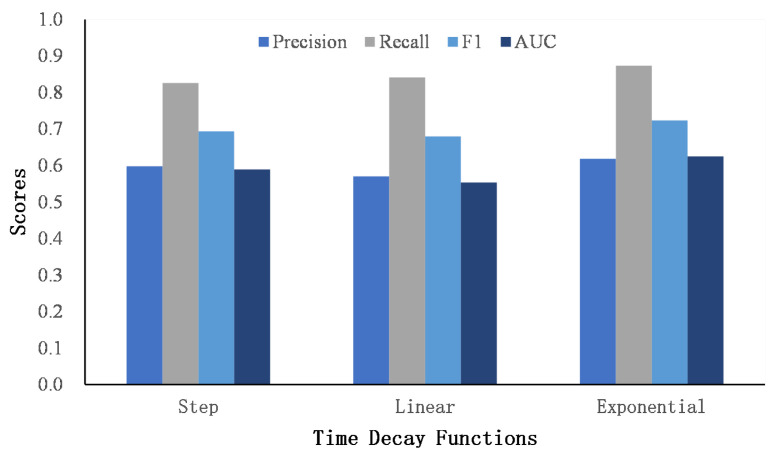
The comparison of time decay function on Flickr–Yelp.

**Figure 7 entropy-24-01603-f007:**
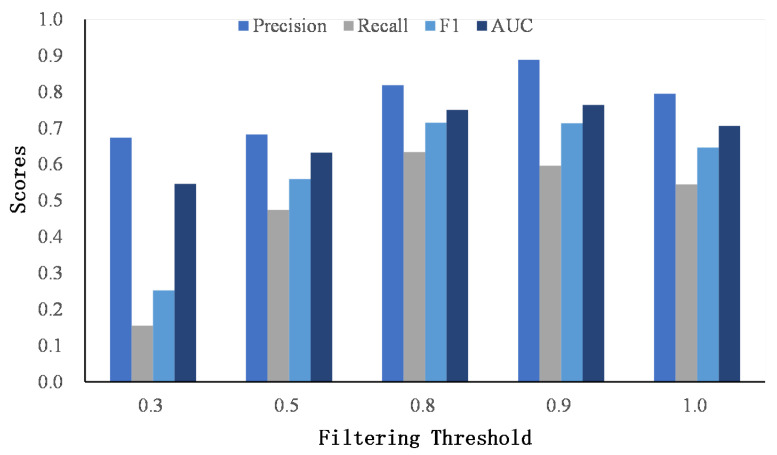
The comparison of the filtering threshold on Foursquare–Twitter.

**Figure 8 entropy-24-01603-f008:**
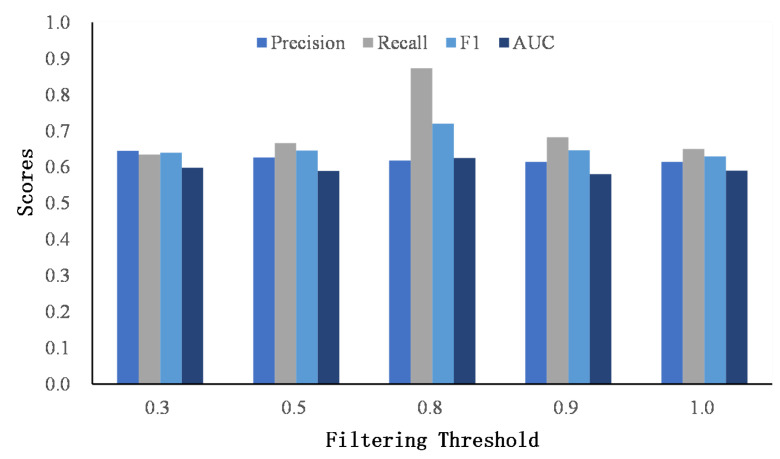
The comparison of the filtering threshold on Flickr–Yelp.

**Figure 9 entropy-24-01603-f009:**
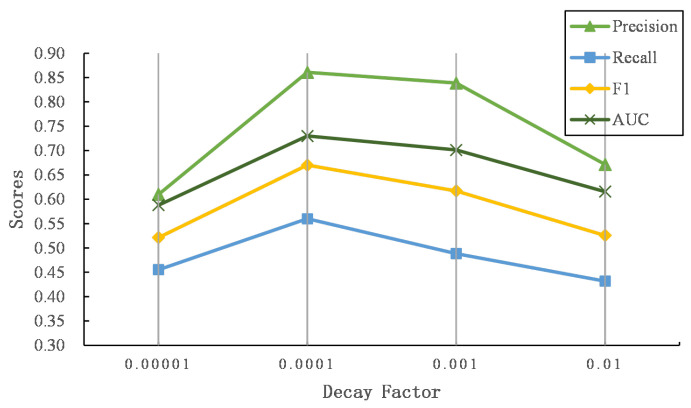
The comparison of the time decay factor on Foursquare–Twitter.

**Figure 10 entropy-24-01603-f010:**
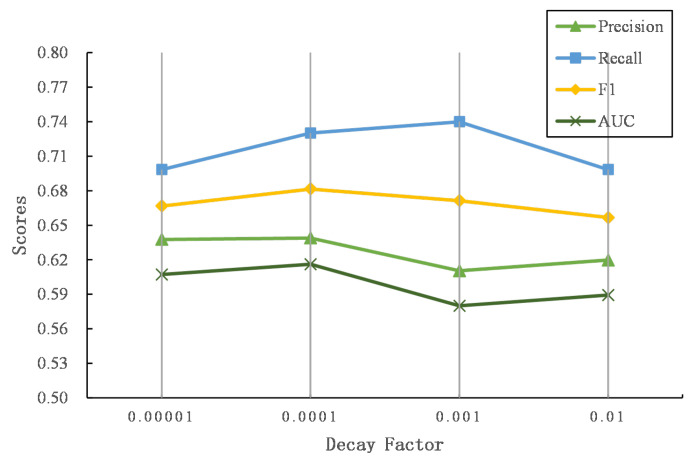
The comparison of the time decay factor on Flickr–Yelp.

**Table 1 entropy-24-01603-t001:** Symbols and definitions.

Notations	Definitions
A,B	Social networks *A* and *B*
$u_{i}^{A},u_{j}^{B}$	Users from social networks *A* and *B*
$C_{i}^{A},C_{j}^{B}$	Content posted by users $u_{i}^{A}$ and $u_{j}^{B}$
$W^{i}=\left\{W_{1}^{i},W_{2}^{i},\ldots ,W_{M}^{i}\right\}$	Words posted by user (taking $u_{i}^{A}$ as an example)
$T_{w}^{i}=\left\{T_{1}^{i},T_{2}^{i},\ldots ,T_{M}^{i}\right\}$	Posting time of words
$Z^{i}=\left\{z_{1}^{i},z_{2}^{i},\ldots ,z_{M}^{i}\right\}$	Embeddings of words
$E^{i}=\left\{E_{1}^{i},E_{2}^{i},\ldots ,E_{P}^{i}\right\}$	Named entities posted by user
$T_{e}^{i}=\left\{T_{1}^{i},T_{2}^{i},\ldots ,T_{P}^{i}\right\}$	Posting time of named entities
$H^{i}=\left\{h_{1}^{i},h_{2}^{i},\ldots ,h_{P}^{i}\right\}$	Embeddings of named entities
$M^{w},M^{e}$	Word/named entity similarity matrix
$\Delta T^{w},\Delta T^{e}$	Time difference matrix of word/named entity
${\hat{M}}^{w},{\hat{M}}^{e}$	Reweighted word/named entity similarity matrix
$x_{w},x_{e}$	Word/named entity matching signals
${\hat{x}}_{w},{\hat{x}}_{e}$	Word/named entity matching signals after attention
*r*	Filtering threshold

**Table 2 entropy-24-01603-t002:** Statistics of datasets.

	Users	Records	Duration
**Foursquare**	5392	48,586	48 months
**Twitter**	5223	8,189,964	72 months
**Flickr**	735	113,966	94 months
**Yelp**	735	16,374	88 months

**Table 3 entropy-24-01603-t003:** Results on Foursquare–Twitter and Flickr–Yelp datasets.

	Methods	Category	Precision	Recall	F1	AUC
**Foursquare–Twitter**	**WAI**	Feature-based	0.5600	0.5049	0.5310	0.5541
	**LSIF**	Feature-based	0.5220	0.5042	0.5129	0.5110
	**UBL**	Topic-based	0.5054	0.6885	0.5829	0.5114
	**DCIM**	Topic-based	0.5278	**0.6976**	0.6009	0.5233
	**Dplink**	Check-in-based	0.6793	0.5972	0.6356	0.7074
	**UGCLink**	Semantic-based	0.8505	0.5729	0.6846	0.7361
	**EKG**	Semantic-based	**0.8881**	0.5962	**0.7134**	**0.7638**
**Flickr–Yelp**	**WAI**	Feature-based	0.5282	0.5102	0.5190	0.5272
	**LSIF**	Feature-based	0.5348	0.6803	0.5988	0.5442
	**UIB**	Topic-based	0.5346	0.7346	0.6189	0.5476
	**DCIM**	Topic-based	0.5250	0.7143	0.6052	0.5332
	**Dplink**	Check-in-based	-	-	-	-
	**UGCLink**	Semantic-based	0.6061	0.7407	0.6667	0.5833
	**EKG**	Semantic-based	**0.6180**	**0.8730**	**0.7237**	**0.6250**

**Table 4 entropy-24-01603-t004:** The comparison of an ablation study on the Foursquare–Twitter dataset. EKG is the original model, EKG-D replaces the time decay function with time discretization, EKG-W removes the named entity modeling, and EKG-E removes the word modeling.

	Precision	Recall	F1	AUC
**EKG**	0.8881	0.5962	0.7134	0.7638
**EKG-D**	0.8453	0.5312	0.6524	0.7170 (↓ 6%)
**EKG-W**	0.8626	0.5305	0.6570	0.7269 (↓ 5%)
**EKG-E**	0.5940	0.6610	0.6257	0.5313 (↓ 30%)

**Table 5 entropy-24-01603-t005:** The comparison of ablation study on the Flickr–Yelp dataset. EKG is the original model, EKG-D replaces the time decay function with time descretization, EKG-W removes the named entity modeling, and EKG-E removes the word modeling.

	Precision	Recall	F1	AUC
**EKG**	0.6180	0.8730	0.7237	0.6250
**EKG-D**	0.5884	0.7186	0.6470	0.5663 (↓ 9%)
**EKG-W**	0.5913	0.8730	0.7051	0.5893 (↓ 6%)
**EKG-E**	0.5471	0.9206	0.6864	0.5267 (↓ 16%)

## Data Availability

The data presented in this study are openly available at https://osf.io/9tzfh/ (accessed on 3 October 2022).

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
