# Peer review of "User Identity Linkage across Social Networks with the Enhancement of Knowledge Graph and Time Decay Function"

_entropy, 2022, doi:10.3390/e24111603_

Round 1
Reviewer 1 Report
The topic of the article is current and very interesting. It refers to exploring the correlations of contents generated by users for predicting user identity linkage. To explore this the Authors propose a user identity linkage model with the Enhancement of Knowledge Graph (EKG), where the continuous time decay function is designed to capture the closeness of posting time. To demonstrate the effectiveness of proposed model, the Authors conduct experiments on two real public datasets.
I have to admit that on the one side the text is thoroughly compiled, but on the other hand is not easy to analyse, because is filled with abbreviations, both semantic and mental. Also data processing methodology is not clearly described.
However, despite these shortcomings, I recommend the article for publication.
Author Response
Dear reviewer,
Thank you very much for the positive comments and constructive suggestions. We have taken them carefully into account in revision and addressed every one of them. The response and revisions are listed below.
Point 1: It is not easy to analyse, because is filled with abbreviations, both semantic and mental.
Response 1: Thank you very much for your kind comment. We have explained the meanings of the abbreviations of the baseline methods in section 5.2 (From line 316 to 354). The abbreviations of the baseline methods come from the titles of these method, and the abbreviations are used in the literature published before. We really hope this will make the meanings of the abbreviations semantically understood. To make the abbreviations clearer, we have replaced the abbreviations of “EKG” with “our proposed method” in section 5.5 (From line 375 to 413) for better understandings. In the ablation study part (Section 5.6), we explain the meanings of the abbreviations of the variance of our model in the caption of the Table 4 and 5 to make it easy to analyse.
Point 2: Data processing methodology is not clearly described
Response 2: Thank you so much for constructive suggestion. We have described the data processing methodology in Section 5.1 (From line 281 to 290) with clearer steps. We describe how to remove the stop words by the package NLTK. Besides, we introduce the lemmatization implemented by WordNet to lemmatize the word. We provide the hyperlinks of the packages used in our experiments.
We sincerely hope that this revised manuscript has addressed all your comments and suggestions. We appreciate for reviewer’s warm work earnestly, and hope that the correction will meet with approval. Once again, thank you very much for your comments and suggestions.
Sincerely,
Hao Gao

Reviewer 2 Report
Thank you for giving me to review your manuscript. This manuscript is interesting and scientifically meaningful for considering identity linkage across social networks. Regarding the contents, the following revision should be considered.
The title should include the study design.
What is the research question of this article? The authors should clarify the gap and the research questions to drive this research.
There are some long paragraphs. The author should focus on theory building, the problems, and research question paragraphs. The first to third paragraphs contain mixed contents. The first paragraph should focus on the general information regarding optimization of primary and secondary control. Moreover, the second and third paragraphs should introduce research question as the theoretical and conceptual framework, including research questions.
The introduction should clearly include the research question of this study.
The sample section of the method contains no descriptions regarding sample calculation.
The discussion part should be based on paragraph writing. There are too long paragraphs, and they are not friendly for readers.
This study should describe the limitation of sampling bias and the results' applicability to other settings, and the future investigation in the limitation part.
In the conclusion or discussion, the study’s strengths should be focused on international readers.
Conclusion should be modest. This study has many limitations and should describe just a possibility.
Author Response
Dear reviewer,
Thank you very much for the helpful comments and constructive suggestions. We have taken them carefully into account in revision and addressed every one of them. The response and revisions are as follows.
Point 1: The title should include the study design.
Response 1: We very much appreciate your kind suggestion. we have updated the title of our study, which is now “User Identity Linkage Across Social Networks with the Enhancement of Knowledge Graph and Time Decay Function”. Compared to the title before, we add “time decay function” in the title, which we designed for avoiding time discretization.
Point 2: What is the research question of this article? The authors should clarify the gap and the research questions to drive this research.
Response 2: Thank you very much for your kind comment. There are two research questions in this article. The first is that modeling the similarity of words is not semantically sufficient for predicting user identity linkage. The reason is that the texts posted by users contain lots of named entities, which are connected in the knowledge graph. Traditional methods ignore the relationships of named entities in the knowledge graph. However, the relationships of named entities supplement the correlations of users across different social networks, which helps to predict whether their identity is consistent or not. The second question is that time discretization will lead to the missing matching in the texts. Traditional methods first divide the texts into time slices based on the posting time, and measure the similarity of texts in each slice. However, some similar texts can be divided into different slices, thus the similarity of these texts is not modeled, leading to the failure of identity linkage prediction. In the manuscript, we illustrate these two problems in Figure 2. These two problems are the gap between traditional methods and our proposed method, which drive our study. We updated the research question in the second paragraph of introduction section to make it clear for understanding.
Point 3: There are some long paragraphs. The author should focus on theory building, the problems, and research question paragraphs. The first to third paragraphs contain mixed contents. The first paragraph should focus on the general information regarding optimization of primary and secondary control. Moreover, the second and third paragraphs should introduce research question as the theoretical and conceptual framework, including research questions.
Response 3: Thank you very much for your constructive suggestion. We totally agree with your suggestion. Therefore, we divide the long paragraphs to small paragraphs for better understandings. We move the description of the Figure 2 in the second paragraph to the caption part to make the research question clearer for readers. To solve the mixed contents problems, we have re-organized the first to the third paragraphs for better understandings. The second paragraph in the old manuscript is the related work, and we move it to the related work part. In the revised manuscript, we introduce the definition and application of user identity linkage tasks in the first paragraph. In the second paragraph, we describe the research question, and the gap between the traditional methods and our methods. In the third paragraph, we introduce the framework of our method and how the method solves the research questions.
Point 4: The introduction should clearly include the research question of this study.
Response 4: Thank you very much for your constructive suggestion. We have updated the research question part of our study in the second graph to make it clearly (From line 40 to 51).
Point 5: The sample section of the method contains no descriptions regarding sample calculation.
Response 5: Thank you very much for your constructive comment. Yes, we have described our sample calculation in the caption of Figure 3. Besides, we add another chapter (Section 4.4, From line 238 to 255) that describes the overall procedures of our model. To make the process of the calculation clearer, we organize the pipeline of model in Algorithm 1 (Line 236).
Point 6: The discussion part should be based on paragraph writing. There are too long paragraphs, and they are not friendly for readers.
Response 6: Thank you very much for your kind suggestion. Yes, we have now divided the paragraphs in the discussion part in the revised version. In section 5.5, we divide the whole paragraph based on the topics discussed in each paragraph. In section 5.7.1 and 5.7.2, we divide the paragraph into two paragraphs, where the settings are introduced in the first paragraph, and analysis is described in the second paragraph.
Point 7: This study should describe the limitation of sampling bias and the results' applicability to other settings, and the future investigation in the limitation part.
Response 7: Thank you very much for your constructive suggestion. We completely agree with your idea. We have described the limitations of our study and the future investigations in Section 6 (From line 490 to 510). There are three major limitations. First, some users on different social networks may not post similar texts in a close time period. Thus, the method modeling the texts posted by users, including our method, may fail. The future investigation will be introducing additional information (such as attribute, check-in information) of users to help to predict identity linkage. Second, we randomly sample the negative users in the experiment. However, in the real world, the number of negative users is more than 1. As a result, it is hard to train the model under the imbalance environments. The future directions can be learning the classifier under the imbalance scenario to improve the performance of the model. Third, the parameters of time decay function (such as time decay factor α in exponential decay function) are set to be the same for all users. However, users may have different habits when posting texts on different social network, and the parameters may not be consistent for all the users. Thus, a future direction can be learning the different parameters of time decay functions for different users based on the users’ habits.
Point 8: In the conclusion or discussion, the study’s strengths should be focused on international readers.
Response 8: Thank you very much for your kind comment. We asked several colleagues who are native English speakers to check the conclusion and discussion. We believe the language is now polished for international readers.
Point 9: Conclusion should be modest. This study has many limitations and should describe just a possibility.
Response 9: Thank you very much for your constructive suggestion. We have described the limitations of our study in the conclusion part (From line 529 to 533), and we describe the future directions of the study.
We sincerely hope that this revised manuscript has addressed all your comments and suggestions. We appreciate for reviewer’s warm work earnestly, and hope that the correction will meet with approval. Once again, thank you very much for your comments and suggestions.
Sincerely,
Hao Gao

Round 2
Reviewer 2 Report
The manuscript has been considerably improved. I think that this paper is suited for inclusion in our journal.
Author Response
Response to Reviewer 2 Comments
Dear reviewer,
Thank you very much again for the helpful comments and constructive suggestions. We have taken them carefully into account in revision and addressed every one of them. The response and revisions are as follows.
Point 1: You tried to do in your rejoinder response 2, but did not include the text in the introduction! Please modify the introduction to provide a clear statement of the question(s).
Response 1: Thank you very much for your kind comment. We have included the research questions explicitly in introduction (From line 52 to 58) to make it clear.
Point 2: Your list of contributions (lines 67 ...) should then relate to your research questions.
Response 2: Thank you very much for your construction comment. We have updated the contributions (From line 76 to 83) that related to our research questions.
Point 3: Your introduction still rambles and has long paragraphs. e.g. you should split para 2 at line 38, starting a fresh paragraph with "Despite the success ..."
Response 3: Thank you very much for your kind comment. We have splited the paragraph 2 to make in clear (Line 39). To reduce long paragraphs, we also split the paragraph 3 into two paragraphs(Line 67).
We sincerely hope that this revised manuscript has addressed all your comments and suggestions. We appreciate for reviewer’s warm work earnestly, and hope that the correction will meet with approval. Once again, thank you very much for your comments and suggestions.
Sincerely,
Hao Gao
